# Generation of *Loa loa* infective larvae by experimental infection of the vector, *Chrysops silacea*

Lontum B. Ndzeshang[1,2], Fanny F. Fombad[1,2], Abdel J. Njouendou[1,2], Valerine C. Chunda [1,2], Narcisse V.T. Gandjui[1,2], Desmond N. Akumtoh[1,2], Patrick W. N. Chounna[1,2], Andrew Steven[3], Nicolas P. Pionnier [3], Laura E. Layland[4,5], Manuel Ritter [4], Achim Hoerauf[4,5], Mark J. Taylor [3], Joseph D. Turner [3]*, Samuel Wanji[1,2]

**1** Parasites and Vector Biology research unit (PAVBRU), Department of Microbiology and Parasitology, University of Buea, Buea, Cameroon, **2** Research Foundation for Tropical Diseases and the Environment (REFOTDE), Buea, Cameroon, **3** Centre for Drugs and Diagnostics, Department of Tropical Disease Biology, Liverpool School of Tropical Medicine, Pembroke Place, Liverpool, United Kingdom, **4** Institute for Medical Microbiology, Immunology, and Parasitology (IMMIP), University Hospital Bonn, Bonn, Germany, **5** German Centre for Infection Research (DZIF), Partner Site, Bonn-Cologne, Bonn, Germany

* Joseph.turner@lstmed.ac.uk

**Data Availability Statement:** All relevant data are within the manuscript.

**Funding:** This work was supported by a Bill & Melinda Gates Foundation Grand Challenges

## Abstract

Basic and translational research on loiasis, a filarial nematode infection of medical importance, is impeded by a lack of suitable *Loa loa* infection models and techniques of obtaining and culturing life cycle stages. We describe the development of a new method for routine production of infective third-stage larvae (L3) of *L. loa* from the natural intermediate arthropod vector host, *Chrysops silacea*, following experimental infection with purified microfilariae. At 14-days post-infection of *C. silacea*, the fly survival rate was 43%. Survival was significantly higher in flies injected with 50 mf (55.2%) than those that received 100 mf (31.0%). However, yield per surviving fly and total yield of L3 was markedly higher in the group of flies inoculated with 100 mf (3474 *vs* 2462 L3 produced). The abdominal segment hosted the highest percentage recovery of L3 (47.7%) followed by head (34.5%) and thorax (17.9%). *L. loa* larval survival was higher than 90% after 30 days of *in vitro* culture. The *in vitro* moulting success rate to the L4 larval stage was 59.1%. After experimental infection of RAG2$^{-/-}$IL-2γc$^{-/—}$ mice, the average *L. loa* juvenile adult worm recovery rate was 10.5% at 62 dpi. More than 87% of the worms were recovered from the muscles and subcutaneous tissues. Worms recovered measured an average 24.3 mm and 11.4 mm in length for females (n = 5) and males (n = 5), respectively. In conclusion, *L. loa* mf injected into *C. silacea* intrathoracically develop into infective larvae that remain viable and infective comparable to L3 obtained through natural feeding on the human host. This technique further advances the development of a full laboratory life cycle of *L. loa* where mf derived from experimentally-infected animals may be utilized to passage life cycle generations via intrathoracic injections of wild-caught vector hosts.

Explorations grant (no. OPP1119043) to J.D.T., S. W., and M.J.T and a grant awarded to M.R., A.H., S.W from the German Research Council (DFG) (Grant DFG RI 3036/1-1) within the "African-German Cooperation Projects in Infectiology." The funders had no role in study design, data collection and analysis, decision to publish, or preparation of the manuscript.

**Competing interests:** The authors have declared that no competing interests exist.

## Author summary

The Neglected Tropical Disease, loiasis (tropical eye worm disease) affects 13–15 million individuals in Central Africa. Loiasis has recently been identified as a cause of premature mortality and is a barrier to ivermectin-based elimination of onchocerciasis or lymphatic filariasis where co-infections occur, due to the risk of serious adverse events. Loiasis lacks laboratory preclinical models for drug development, biomarker discovery and inflammatory pathology research. Here we detail the successful development of an experimental technique for the laboratory production of *Loa loa* infective larvae via injection of purified microfilariae into the thorax of wild-caught tabanid flies that are the natural transmission vector. The high yielding infective larvae produced in the laboratory were validated as biologically viable in culture and in a new mouse infection model whereby adult-stage parasites could be produced. The evidence reported herein is an important step to establishing a full laboratory life cycle of *L. loa* by passage between animal models and experimental injections of the wild-caught vector, *Chrysops*.

## Introduction

Loiasis is a vector-borne, neglected tropical disease (NTD), caused by the tissue dwelling nematode parasite, *Loa loa* and transmitted by the tabinid flies *Chrysops silacea* and *C. dimidiata* [1, 2]. Geographically, the disease is limited by the distribution of its two biting tabanid vectors and is endemic in 11 Western and Central African countries particularly in forested areas [3]. It is estimated that some 14.4 million people live in high risk areas where the prevalence of loiasis is greater than 40%, with 15.2 million in intermediate risk areas where the estimated eye worm prevalence is between 20 and 40% [3]. The common clinical signs of loiasis are migration of the adult worm, Calabar swelling, pruritis and arthralgia [4]. Loiasis is a neglected but nevertheless epidemiologically relevant tropical disease which has, until latterly, attracted only limited attention in drug or vaccine research and development. In the past two decades, interest in this disease has increased due to reports of severe adverse events (SAEs) post-ivermectin during mass treatment of the priority NTD, onchocerciasis (river blindness) in areas of onchocerciasis / loiasis co-endemicity. This situation has hampered the mass drug administration (MDA) with ivermectin for the elimination of onchocerciasis in areas where *L. loa* prevalence exceeds 20% [5, 6]. Therefore, this has impeded the goals stated by the World Health Organization, to eliminate onchocerciasis as a public health problem by 2030. This includes the permanent protection of the remaining 120 million people at risk of this debilitating and disfiguring disease in 19 countries in Africa through the establishment of community-directed treatment with ivermectin (CDTi), in areas of co-endemicity. Presently, there is no satisfactory treatment for loiasis though two medications have so far been employed for clinical cases namely, diethylcarbamazine (DEC) and albendazole. Several cases of coma and even death have been reported in individuals with high microfilarial (mf) loads of *L. loa* when treated with DEC [7]. Whilst albendazole-based clinical trials have shown promise in lowering *Loa* microfilaraemias, more precise dose optimisations are required to define a minimally sufficient and safe dosing regimen. If achievable, albendazole short-courses or other, new, safe anti-loiasis drugs could be administered before MDA with ivermectin in areas of co-endemicity [8, 9]. Loiasis drug discovery and development necessitates the convenient and abundant availability of parasite material suitable for screening compounds both *in vitro* and *in vivo*. While the mf stages can be purified from the blood of infected humans or primates, the production of infective larvae requires infection of the *Chrysops* vectors with the mf. Several methods have been

used to produce infective larvae of filariae from flies and include recovery of L3 from engorged flies on consenting microfilaraemic donors (human landing catch) [10–12], baited traps (wood fire) and multiple fly dissections from hyperendemic hot-spots of infection and artificial membrane feeding systems [13, 14]. However, applying the latter technique for infection of *Chrysops* for L3 production have been unsuccessful as these flies are refractory to feeding in captivity. Bianco *et al.* [15], Dohnal *et al.* [16] and Fukuda *et al.* [17] produced infective larvae of *Onchocerca* species by experimentally infecting *Simulium* blackfly intrathoracically with mf. Intrathoracic injections of flies with mf bypass feeding of flies on donors, and therefore address the ethical concerns of human landing catch. The use of animal sources of mf, such as splenectomised baboons [11] or immunodeficient mice [18] would potentially completely obviate the use of human donors and may achieve a complete laboratory life cycle local to a source of wild-caught vectors. Further, high yielding production of L3 following experimental infections of *Chrysops* would be a significant improvement in throughput for translational medicine applications compared to standard methods. In this study, we assessed if intrathoracic *L. loa* mf injected *C. silacea* can survive under laboratory conditions for a minimum of two weeks and whether L3 infective larvae recovered from lab maintained flies could maintain their viability and development both *in vitro* and *in vivo* taking advantage of recent advances in *L. loa* culture systems and mouse models developed by our laboratories [12, 18].

## Methods

### Ethics statement

The study design and protocols were approved by the REFOTDE Institutional Animal Ethics Committee (RIAEC), and ethical clearances (001/RIAEC/2015) were issued. Animal experimentation were in strict accordance with the international guidelines of rearing animals and use in medical research, the Animal Welfare Legislation and Policies and complied with the Animals (Scientific Procedures) Act 1986 (ASPA) and its associated codes of practice on animal housing and care [19]. Previous works that used the same procedures are found here [18].

### Experimental animals

RAG2$^{-/-}$IL-2γc$^{-/-}$ C57BL/6 mice used in the experiment were obtained from the Institute of Medical Microbiology, Immunology and Parasitology (IMMIP), University of Bonn, Germany. Mice were shipped in filter topped boxes to Research Foundation for Tropical Diseases and the Environment (REFOTDE), Buea, Cameroon. All mice were reared at the REFOTDE animal facility in a 12:12 light: dark cycle, maintained in individually ventilated caging (IVC) with HEPA filtered air system (Techniplast) with *ad libitum* provision of standard irradiated rodent chow and bottled mineral water. All mice used for the experiments were infected in the laboratory of the REFOTDE. All mouse procedures received ethical approval by the REFOTDE Institutional Ethics Animal Committee (RIEAC) and were undertaken in accordance with UK standards for use of animals in research.

   *Loa loa* microfilariae (mf) were obtained from splenectomised infected baboons (*Papio anubis*) that were kept in captivity and infected with the *L. loa* human strain [11]. The acquisition, care and ethical concerns on the use of baboons as donors of mf have been previously documented [10, 11]. Ethical and administrative approvals for the use of baboons in this study were obtained from the Ministry of Scientific Research and Innovation of Cameroon (Research permit #028/ MINRESI/ B00/ C00/ C10/ C12) and the RIEAC. Procedures adhered to the NIH Guide for the Care and Use of Laboratory Animals.

## Collection of wild *C. silacea*

Wild *C. silacea* (with an infection rate of less than 1%) were collected from Kindongi rainforest reserve [1] located in the South-West Region of Cameroon. Catches were made from 8 a. m. to 4 p.m. during the mild rainy and mild dry seasons especially during sunny hours. *C. silacea* is the only vector species found in this study site [1]. Flies were attracted by the smoke from wood fire, then caught using sweep nets (with multiple fly dissections to control for any naturally acquired infection) and transferred to individually labelled 50 mL Falcon tubes (Corning, USA) prepared to provide suitable conditions for their survival during their transportation to the laboratory in a cold box as described by Wanji *et al.* [11].

## Isolation and purification of *L. loa* mf from blood

*L. loa* mf were obtained from experimentally infected baboons (*Papio anubis*) with the human strain of *L. loa*. The infection and follow up of these animals were previously described [11]. Approximately 8–10 mL venous blood was collected from microfilaraemic baboons around 9 am into 10 ml EDTA coated tubes (Corning, USA) and transported within 1 hour to the laboratory for processing.

 *L. loa* mf were extracted using Percoll gradient centrifugation as described by Cesbron *et al.* [20] with modifications [21]. Briefly, an aliquot of 2.5 ml of baboon whole blood was pipetted and layered gently onto a Percoll (GE Healthcare, Pharmacia, Uppsala, Sweden) gradient (40,50 and 65%) in a 15 ml centrifuge tube (Corning, Kennebunk-ME, USA) and centrifuged (Humax 14k human, Wiesbaden, Germany) at 3000 rpm for 10 minutes at 25°C. The layer of Percoll gradient containing microfilariae was removed using a syringe and pumped gently through a 5 μm pore size cellulose filter into a 50 ml falcon tube. The syringe was carefully removed and 10 ml of RPMI-1640 (Sigma, USA) was aspirated into the syringe to wash the filter. The filter was transferred immediately to a Petri dish containing culture medium and incubated at 37°C for 5 minutes. The filter was then removed from the Petri dish and the suspension centrifuged at 1500 rpm for 10 minutes to concentrate the parasites which were then quantified using an inverted microscope. Concentrations of 50 mf and 100 mf load/ 15 μl of medium were prepared as previously described [22].

## Injection of *C. silacea* with *L. loa* mf

Prior to injection, the flies were knocked down by placing the 50 ml Falcon tube containing the flies at -20°C for 5 minutes. Each fly was removed from the tube, orientated in a supine aspect, and held in place with sterile forceps on a cold light source platform of a dissecting microscope. The fly was then injected at the left side through the blind spot of the thoracic region with a 15μl of medium containing 50 (group 1) or 100 (group 2) mf using a 0.5 ml insulin syringe (Fig 1). After injection, flies were placed into individual rearing tubes and monitored for survival.

## Laboratory maintenance of infected *C. silacea*

Flies were transferred to the insectarium and monitored daily for 14 days (intrinsic incubation period for mf development to L3 stage). During this period, flies were fed daily with a sterile 15% sucrose solution. The temperature of the insectarium was maintained between 21–23°C with 79% - 80% relative humidity [10]. Fly mortality was recorded daily to determine the survival rate.

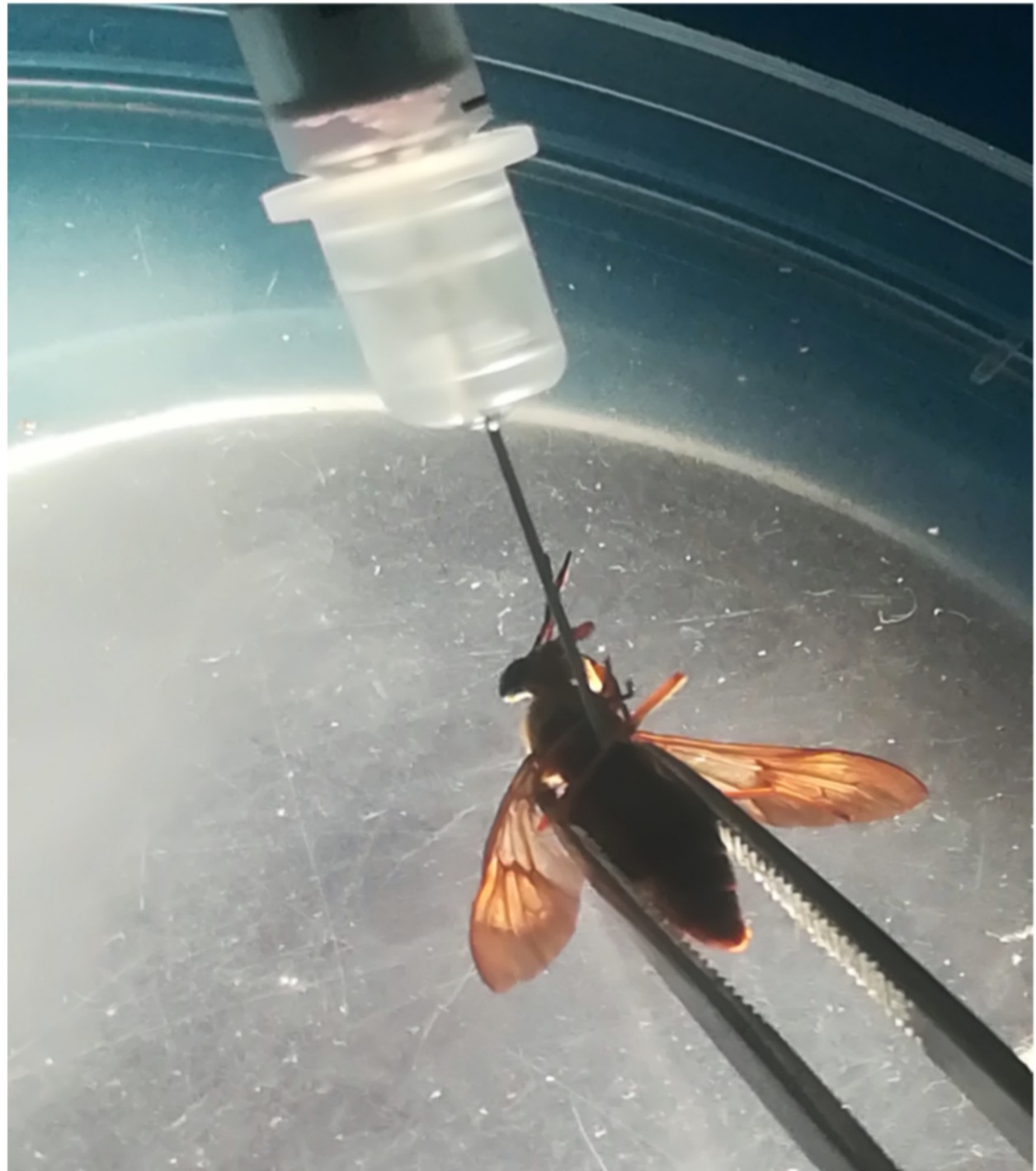

**Fig 1. Injection of *Loa loa* mf into *C. silacea*.** Here, 0.5 ml insulin syringe containing *Loa loa* mf is introduced to a ventral positioned *Chrysops* fly held with a pair of forceps.

### Dissection of flies for *L. loa* L3 recovery

At 14 days post infection (p.i.), flies were placed in sterile distilled water containing 0.2% Tween 20 (Sigma-Aldrich, St Louis, USA) in a sterile Petri dish for 1 min, transferred to a cleaning solution (sterile distilled water) and placed into a Petri dish of diameter 5.5 cm containing dissecting medium (RPMI-1640 medium; Sigma-Aldrich, St Louis, USA) using sterile forceps. The head, thorax and abdomen were separated from each other, placed in Petri dishes and dissected with sterile needles under a dissecting microscope (Motic, Wetzlar, Germany). Emerging larvae were collected and transferred with a sterile micropipette into the first 3 wells of a depression or concentration plate filled with sterile RPMI 1640 medium. Infective larvae from head, thorax and abdomen were recovered and processed separately. The larvae were washed 3 times by transferring into different depression wells. Isolated parasites were quantified, and motility/survival rates assessed using an inverted microscope (Motic AE 21 series, Wetzlar, Germany).

### *In vitro* culture

Infectious L3 larvae were transferred into centrifuge tubes containing dissection medium using a sterile pipette. The infective larvae were purified by a Percoll gradient technique [21]. Briefly, L3 were centrifuged (Centrifuge; HUMAX 14K HUMAN, Germany) at 1500 rpm for 10 mins and later concentrated in less than 1 ml RPMI. The pellet was slowly layered on the surface of a 15 ml tube containing stock iso-osmotic Percoll (GE Healthcare, Pharmacia, Uppsala, Sweden) and centrifuged at 800 rpm for 10 minutes. The process was repeated to reduce contaminants. At the end, the L3 were washed twice with RPMI-1640 by centrifugation at 1500 rpm for 10 minutes to remove residual Percoll. The L3 were then cultured in Dulbecco's Modified Eagle Medium (DMEM; Gibco Life Technologies, Cergy-Pontoise, France) supplemented with 10% foetal bovine serum (FBS; Lonza Verviers, Belgium) on monkey kidney epithelial cells (LLC-MK2; ATCC, Virginia, USA) in a flat bottom 48 well plates (Corning, Kennebunk-ME, USA) containing lids for a maximum period of 30 days at 37˚ C, 5% $CO_2$ in an incubator ($CO_2$ series Sheldon Mfg. Inch, Cornelius, OR, USA) while monitoring for parasite motility, survival and moulting [12]. Parasite viability during culture was assessed daily by visual inspection under an inverted microscope for 30 days. Their motility was scored on a 4 point scale: 0; no movement or immotile, 1; intermittent shaking of head and tail, 2; sluggish (shaking of the whole worm on a spot), 3; vigorous movement (shaking of the whole worm and migration from one spot to another) as previously described [12, 22].

### Experimental mouse infection

The L3 larvae were washed and quantified under a dissecting microscope and concentrated in approximately 100–200 μL of RPMI in a 0.5 ml insulin syringe. We have recently established mouse models of loiasis utilising Severe-combined / $\gamma c^{-/-}$ compound immunodeficient mouse strains [18]. We experimentally infected groups of 5–6 mice subcutaneously in the lumbar area [10] with L3 (ranging from 50–140) derived from the head or abdomen of experimentally-injected *C. silacea*. Due to limited numbers of L3 recovered from the thorax of *C. silacea*, a single mouse was infected and an additional five mice were infected with a mixture of L3 derived from the thorax and abdomen. Infected animals were euthanized by a rising concentration of $CO_2$ and cervical dislocation to confirm death before dissection 62 days p.i. for parasite recovery. The protocol used in the dissection of mice was described in previous studies [18, 24]. Initially, blood was collected by cardiac puncture and two slides of 50 μL each prepared (stained with Giemsa and viewed under a microscope). This was done to check for the absence of *L. loa* mf [11]. Animals were dissected, and the different tissues were gently teased

out and incubated in RPMI medium to allow for free migration of larvae into the medium as previously described [10, 23]. Specifically, the skin (subcutaneous tissues), muscles, peritoneal and pleural cavities, heart and lung tissues were incubated in separate petri dishes. The larvae recovered were counted and the site of recovery noted. The larvae recovered were stored in 80% ethanol for measurement of their lengths and widths. For morphological observations, 5 female and 5 male worms were treated with Amann lactophenol and viewed under a light microscope as described by Ushirogawa *et al.* [24] and Bain *et al.* [25].

### Statistical analysis

The survival rates of the flies at each time point was estimated as the percentage of the remaining number of flies alive for each batch. Mann-Whitney-U-Test and Chi square test were used to compare the survival rate (%) at day 14 between the two fly groups. For L3 recovery, test for normality was done using Anderson-Darling test and L3 recovery between groups was compared using independent-Samples median test. The L3 recovery from different fly anatomical locations were compared using Friedman test (Head vs Thorax vs Abdomen) and Wilcoxon Signed Rank test to compare pairwise (Head vs Thorax, Head vs Abdomen and Thorax vs Abdomen). Mean motility and mean survival were computed to assess the viability of the parasites *in vitro*.

Cuzick test for trend was used to compare the moulting rate of *L. loa* L3 from different parts of the *C. silacea*. The proportions of larvae recovered from each region of the mice and mean number (%) of larvae recovered (with respect to provenance of L3) were compared using the Chi-square and Kruskal-Wallis tests. All the tests were performed at 5% level of significance. Statistical analysis was undertaken using SPSS version 20, GraphPad Prism 8.0.2, and the *PMCMRplus* package in R version 3.6.1.

### Results

### *C. silacea* experimentally injected with *L. loa* mf survive and produced L3 larvae

Survival of flies injected with either 50 or 100 purified *L. loa* mf after 14 days are detailed in Table 1 and Additional flies 1 and 2. Out of 428 *C. silacea* injected with *L. loa* mf, 184 survived (42.0%) and were dissected for L3 recovery. At day 14 p.i., group 1 (50 mf inoculated) recorded a higher survival rate compared to group 2 (100 mf inoculated) (55.2% *vs* 31.0% survival, $\chi^2 = 18.1$, $P<0.001$)

A total of 184 *C. silacea* were dissected and the recovered L3 were quantified. At least 2 infective L3 were recovered from each dissected fly. The median L3 recovered and parasite yields per injected *Chrysops* fly were calculated (Table 1). The median L3 recovered per surviving fly in group 2 was significantly greater than in group 1 (56 *vs* 19, P < 0.001, independent

**Table 1.** *C. silacea* survival rate and number of L3 of *L. loa* recovered 14 days following intrathoracic injection of mf.

| Fly group | mf load injected / fly | No. flies injected | No. of flies surviving, day 14 | Fly survival rate, day 14 (%) | Percentage of *Chrysops* with L3 (%) | Median L3 recovered / fly | Median L3 yield / fly (%) | Total L3 recovered |
|---|---|---|---|---|---|---|---|---|
| 1 | 50 | 212 | 117 | 55.19* | 100 | 19 | 42.08 | 2462 |
| 2 | 100 | 216 | 67 | 31.01 | 100 | 56* | 51.84 | 3473 |

*$\chi^2$ = 18.12 P<0.001 proportion survival of flies between groups, *Independent-Samples median test P<0.001 (median L3 recovered/fly). All surviving flies were dissected for L3 recovery

**Table 2. Average L3 recovered from different parts of *C. silacea* injected intrathoracically with 50 and 100 mf of *L. loa* at day 14 p.i.**

| Group | Number of flies dissected | % L3 recovered from head (mean±SD) | % L3 recovered from thorax (mean±SD) | (%) L3 recovered from abdomen (mean±SD) |
|---|---|---|---|---|
| 1 | 117 | 36.1 (7.6±9.8) | 16.7 (3.5±3.4) | 47.2 (9.94±7.0) |
| 2 | 67 | 33.3 (17.2±21.0) | 18.7 (9.7±12.1) | 48.0 (24.87 ±21.43) |
| Total | 184 | 34.5^ (11.1±15.5) | 17.9 (5.8±8.3) | 47.7*#‡ (15.37 ±15.77) |

p-values for comparison of recovery rate of *L. loa* L3 from different sections of the *C. silacea* body:

*Head vs thorax vs abdomen: P <0.001; Friedman's test

^Head vs thorax: P<0.001; Wilcoxon signed rank test

#Head vs abdomen: P = 0.012; Wilcoxon signed rank test

‡Thorax vs abdomen: P<0.001; Wilcoxon signed rank test

samples median test). Despite the significantly higher mortality rate in group 2, a higher total yield of L3 was produced from surviving flies in group 2 compared with group 1 (3473 *vs* 2462).

The highest recovery percentage of infective larvae was from the abdomen (47.7%) followed by the head (34.5%) and the thorax (17.9%). Variation in percentage recovery between the various fly segments was statistically significant (P<0.001, Friedman Test, Table 2). Subsequent post-hoc testing, comparing average recovery (%) of infective larvae between fly segments, determined that L3 recovery was significantly highest in the abdomen compared with either thorax or head (*P*<0.001 and *P* = 0.012, respectively, Wilcoxon Signed Rank Tests) but that yields in the head were also significantly higher than the thorax (*P* = 0.001, Wilcoxon Signed Rank Test). The distribution of L3 in the three anatomical locations was not different in flies receiving 50 or 100 injected mf.

## *L. loa* L3 generated from mf injections of *C. silacea* develop to fourth-stage larvae and survive within long-term *in vitro* cultures

We have recently developed a long-term mammalian cell co-culture system whereby *L. loa* L3 isolated from wild-caught *C. silacea* remain ≥90% viable for 18 days, with growth and 20–40% moulting success rate to the L4 stage [12]. We exploited these defined *in vitro* culture conditions to evaluate the *in vitro* fitness of experimentally-derived *L. loa* L3. Survival and average motility scores of developing *L. loa* larvae in culture are plotted in Fig 2. The survival of larvae

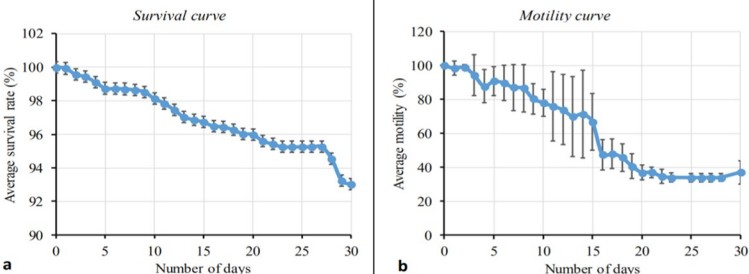

**Fig 2. Survival and motility monitoring of *L.** loa L3 derived from mf injections of *C. silacea*, in mammalian *in vitro* culture conditions. Percentage survival curve (A) and motility curve (B) of *Loa loa* larvae during 30 days of culture. Plotted is the average (mean) response. Bars indicate standard deviations (SD).

remained >90% throughout the course of the experiment (30 days). At day 5, motility was greater than 80% and decreased to 36% (day 20) and stayed constant.

Of the 2462 L3 recovered from flies that were injected with 50 mf, 2453 were successfully placed into culture. Subsequently, 1450/2453 larvae underwent moulting (from L3 stage to L4 stage; Fig 3 & Table 3). Whilst there was a trend for higher moulting rate in larvae derived from the head (61.2%) followed by 60.1% and 56.4% for those derived from the thorax and abdominal segments, respectively (Table 3), this did not attain significance (P = 0.975, Cuzick trend test).

### *L. loa* L3 generated from mf injections of *C. silacea* can successfully infect immunodeficient mice and develop into adult filariae

All worms recovered from mice 62dpi were young adults (Fig 4). The average length and width of female worms recovered was 24.6 mm and 0.35 mm respectively and 11.4 mm and 0.27 mm for male worms (n = 5 worms measured per sex).

Worms were recovered from the subcutaneous tissues, muscles, peritonea and pleural cavities (Table 4 and Additional file 3). Overall, more than 87% of worms were recovered from the muscles and subcutaneous tissues, the natural niche of adults in humans and primates [2, 11]. The proportions of worms recovered from the different anatomical regions of mice were therefore significantly different ($\chi^2$ = 33.8, P<0.001). The provenance of L3 from experimentally

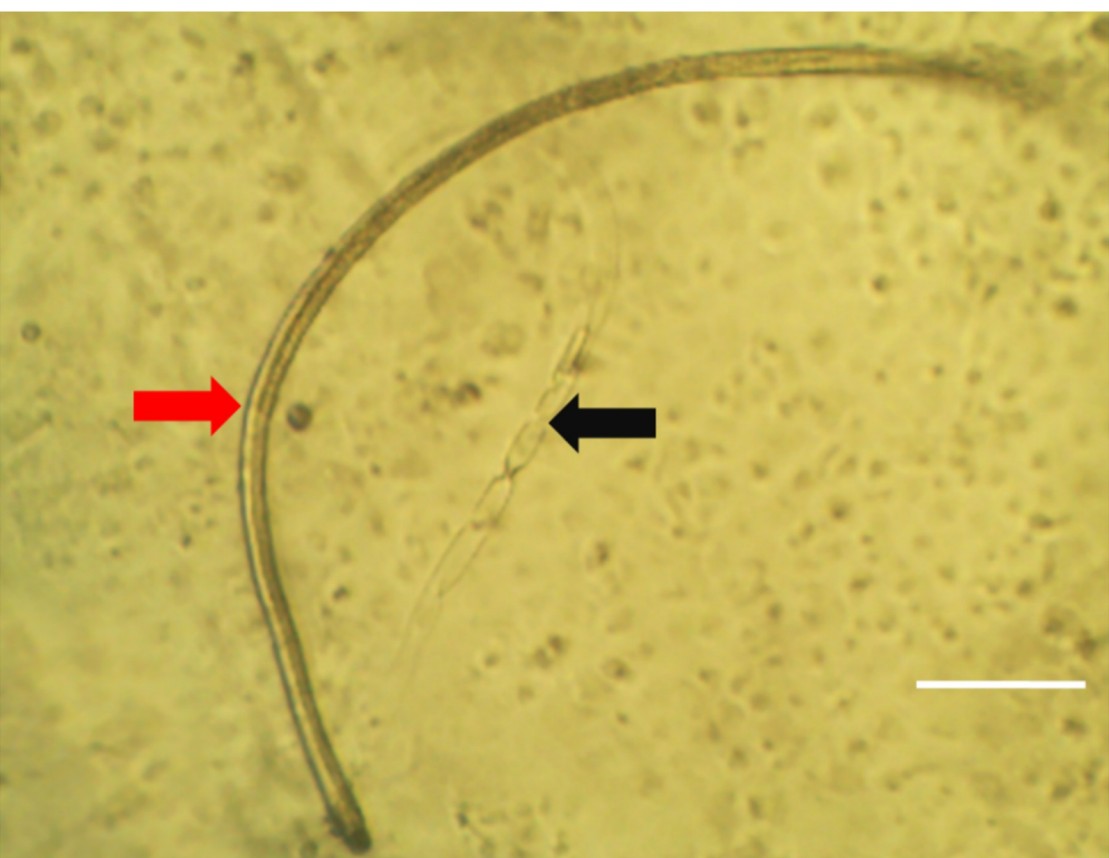

**Fig 3. Representative image of an *in vitro L. loa* L4 larvae and cuticle cast.** Image taken by inverted tissue culture microscope. Arrows: red = fully moulted *Loa* L4; black = Cast cuticle from moulted *Loa* L3. Scale bar 100μM.

**Table 3. *in vitro* culture moulting success of *L. loa* L3 derived from mf injection of *C. silacea*.**

| Parameters | Head | Thorax | Abdomen | Total |
|---|---|---|---|---|
| n L3 plated | 1206 | 502 | 745 | 2453 |
| n L3 moulted | 738 | 297 | 415 | 1450 |
| % moulted | 61.2 | 59.2 | 55.7 | 59.1 |
| Statistics | Cuzick trend test: $z = 0.0311$, $p = 0.975$, | | | |

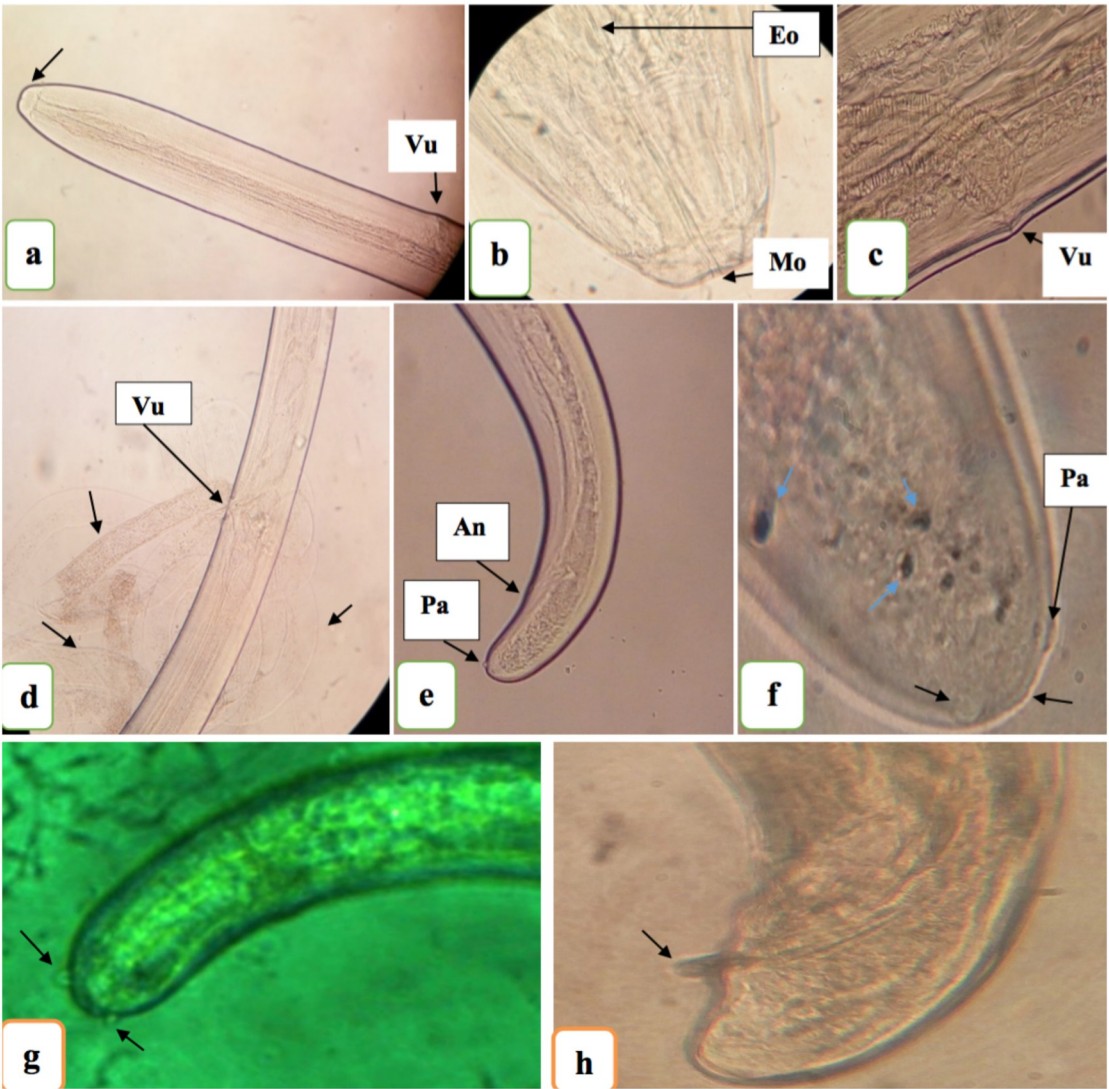

**Fig 4. *Loa loa* adult morphology 62 days post-inoculation.** (A and B) Head of young adult female stage (anterior region); **Mo**–mouth opening; **Eo**—esophagus. (c): Vulva [**Vu**] of mature young adult female (lateral view). (D): Open vulva, intestine, uterus of female young adult worm with no mf present (indicating that the worm is not fully developed [lateral view]). (E and F): Posterior extremity of matured young adult female worm (E; lateral view [**An**–anus] and F; ventral view) with caudal papillae [**Pa**]. (G and H) Caudal region (lateral view) of *Loa loa* young adult male with caudal papillae (G) and spicule (H). [A, D, E, G viewed under x10 objective, B, C viewed under x40 objective F, H viewed under x100 objective].

**Table 4. Recovery of adult *L. loa* from RAG2$^{-/-}$IL-2$\gamma$c$^{-/-}$ mice infected with L3 derived from experimental injections of *C. silacea*.**

| L3 Source | Total L3 inoculated | Mice N | Site of parasite recovery (% total recovered) | | | | Total adult *L. loa* recovered (% L3 inoculate) | Median adult *L. loa* recovererd / mouse % L3 inoculate (IR)* |
|---|---|---|---|---|---|---|---|---|
| | | | Sub-Cut. | muscle | Peritonea | Pleura | | |
| Head | 489 | 5 | 28 (33.3) | 44 (52.4) | 3 (3.6) | 9 (10.7) | 84 (17.2) | 20.8 (1.8–22.2) |
| Abdomen | 565 | 6 | 21 (45.7) | 19 (41.3) | 5 (10.9) | 1 (2.2) | 46 (8.1) | 7.5 (5.5–10.1) |
| Thorax | 123 | 1 | 2 (16.7) | 9 (75) | 1 (8.3) | 0 (0) | 12 (9.8) | 9.8 |
| Thorax + Abdomen | 472 | 6 | 15 (48.4) | 15 (48.4) | 1 (3.2) | 0 (0) | 31 (6.6) | 4.7 (2.2–10.9) |
| All tissues | 1649 | 18 | 66* (38.2) | 87 (50.3) | 10 (5.8) | 10 (5.8) | 173 (10.5) | 7.5 (3.3–12.9) |

Infective larvae of 50–140 were inoculated per mice depending on recovery from fly anatomical regions

*$X^2$ = 33.8, P<0.001, proportion of adults parasitizing specific tissues

infected *C. silacea*, in terms of head, thorax and/or abdomen, did not significantly affect the recovery rate of adult *L. loa* worms from mice 62 dpi (Kruskal Wallis test P = 0.42), although there was a trend for higher recoveries when L3 derived from the head of the vector were used.

## Discussion

In this study, we establish, for the first time, that *L. loa* mf purified from the blood of an experimental baboon host can be successfully injected into wild-caught, laboratory-maintained *C. silacea* to yield viable infectious stage L3 larvae. The yields, quality and infectivity of the infective larvae produced were confirmed via novel *in vitro* and *in vivo* systems recently developed by our laboratories to be comparable to *L. loa* L3 derived from *C. silacea* human landing catches [12] or wood-smoke baited traps [18].

We scrutinised two microfilaraemic doses because we predicted a trade-off between numbers of L3 produced and fly mortality due to high level *L. loa* infection and collateral damage to fly tissues as *L. loa* migrate and develop from the point of injection. Despite the higher mortality rate of flies injected with 100 mf, we determined that *C. silacea* surviving to day 14 yielded significantly higher numbers of L3 per batch compared with flies inoculated with 50 mf. The higher mortality in 100 versus 50 mf injections indicates that infectious dose is the major factor affecting survival of the flies. However, sham infections need to be performed in future to determine what impact the injection technique has on fly survival. Because of the reduced labour time required to dissect the fewer surviving, high-yielding flies, we conclude that 100 mf injections are most suitable for the production of *L. loa* L3.

We evaluated the biological fitness of L3 derived from experimental inoculations of *C. silacea* in three ways: 1. The ability to shed their cuticles (moult) to develop to the L4 stage 2. Their survival in culture over 30 days and 3. Their ability to infect and establish adult infections in susceptible immunodeficient mice after two months. In foetal calf serum supplemented DMEM medium and incubation at standard human tissue culture conditions, we have prior determined that between 20–40% of L3 derived from human-fed *C. silacea* complete the L3-L4 moult and survival, assessed by motility, continues in >90% of cultured larvae for 18 days [12]. In the experiments reported here for experimentally-derived L3, 59.1% completed

the L3-L4 moult and >90% survival was recorded until the end of the experiment at day 30. In *L. loa*, the moulting of L3 to L4 stage occurs near the end of the first week of infection / incubation [25]. Bain and colleagues reported moults of L3 to L4 stage of *L. loa* at day 8 after inoculating infective larvae of *L. loa* recovered from *Chrysops* into mice, whilst our prior *in vitro* findings moulting started from day 9 [12]. Here we observed moulting of L3 to L4 stage was characterised by marked decrease in the activity of the larvae, that was observed as a drop in motility which stabilized at day 23 and a slight increase in motility observed towards day 30 when the majority of larvae had completed their moult.

Our *in vitro* observations suggest that *in vitro* 'fitness' of experimentally derived L3 is at least comparable, if not superior, to L3 derived from naturally-fed flies caught after landing on infected human volunteers and subsequently maintained in our laboratory insectary. Whilst several direct individual experimental comparisons would be required to determine the superior biological fitness of experimentally-reared *L. loa* L3, the data supports the use of experimentally-derived L3 in drug screening or other biological *in vitro* studies.

Utilising our recently established *L. loa* mouse model system [18] immature adult-stage worms were consistently recovered from RAG2$^{-/-}\gamma$c$^{-/-}$ mice at day 62 post-inoculation, with a 10.5% yield of the initial unit inoculum, predominantly from the muscle and subcutaneous tissues; the natural *L. loa* adult parasite niche. The immature adult *L. loa* stage was verified by identification of sex-specific organs and morphological characteristics, as well as the macroscopic size of the filariae recovered. These findings emulate our studies where experimental infections of immunodeficient mice were undertaken utilising L3 from blood fed *Chrysops* both in terms of the matching predominant anatomical locations of adult infections and the yields evident [10, 18]. Following infections of both baboons and immunodeficient mice, the development of patent infections, with mf production seeding the blood, are detectable at between 5 and 6 months [11,18]. Given comparable yields and presence of both male and female immature worms in mice infected with experimentally generated L3, we would predict that patent infections would be possible, and this is the focus of ongoing work. If patent infections can be established, the successful derivation of L3 via experimental injections of mf into *C. silacea* could be utilised to sequentially passage *L. loa* between mice and wild-caught flies to establish a laboratory experimental life-cycle of *L. loa*.

We determined that at day 14 post-injection, the abdominal fly segment recorded the highest proportion of L3, followed by L3 isolated from the head with a minority of L3 derived from the thorax. The abdominal fat bodies serve as the site of mf-L1-L2-L3 development in *Chrysops* [26]. Following mf development to L3 in the abdomen, L3 migrate to the head via the thorax and only the L3 in the head are positioned for transmission when infected flies take a subsequent blood meal [27]. Because it is speculated that L3 derived from the head might be more biologically adapted to successfully transition to mammalian host conditions, we scrutinised whether biological fitness of L3 to develop to L4 *in vitro* or immature adults *in vivo* was associated with the fly anatomical segments they were isolated from. We observed a trend for both increased moulting rate *in vitro* and increased immature adult *L. loa* yields *in vivo* when utilising 'infective' L3 derived from the head of *C. silacea*. However, the majority of L3 derived from the thorax and abdomen could also undergo moulting and L3 isolated from these fly segments could also develop to adult infections in mice. The differences in moulting and adult establishment were not statistically different between L3 isolated from the three fly segments. Our results support the use of pooled L3 derived from all segments of *C. silacea* in onward biological and translational medicine research such as *in vitro* and *in vivo* drug studies. This has the advantage of significantly increasing yields of L3 from experimental injections for onward applications.

## Conclusions

This study is the first demonstrating the production of *L. loa* infective L3 larvae from *C. silacea* inoculated with *L. loa* mf through intrathoracic injections. The quality of infective *L. loa* larvae produced using this method were confirmed in *in vitro* and *in vivo* experimental systems. This technique creates a new platform for the generation of high yields of *L. loa* L3 to increase throughput for *in vitro* and *in vivo* drug screening experiments. We also conclude that via iterative experimental injections of *C. silacea* and passage in mice to provide patent infections, a complete *L. loa* laboratory life cycle is now achievable utilising wild-caught flies. These findings obviate the necessity for human volunteers and non-human primates to generate *L. loa* life-cycle stages and will accelerate translational medical research on this Neglected Tropical Disease by providing a more abundant source of parasitic material to researchers in need.

## Acknowledgments

We thank the local communities for allowing access and assisting in the trapping of *C. silacea*.

## Author Contributions

**Conceptualization:** Joseph D. Turner, Samuel Wanji.

**Formal analysis:** Lontum B. Ndzeshang, Fanny F. Fombad, Abdel J. Njouendou, Valerine C. Chunda, Narcisse V.T. Gandjui, Desmond N. Akumtoh, Patrick W.N. Chounna, Manuel Ritter.

**Funding acquisition:** Manuel Ritter, Achim Hoerauf, Mark J. Taylor, Joseph D. Turner, Samuel Wanji.

**Investigation:** Lontum B. Ndzeshang, Fanny F. Fombad, Abdel J. Njouendou, Valerine C. Chunda, Narcisse V.T. Gandjui, Desmond N. Akumtoh, Patrick W.N. Chounna, Manuel Ritter.

**Methodology:** Andrew Steven, Nicolas P. Pionnier.

**Project administration:** Samuel Wanji.

**Resources:** Laura E. Layland.

**Supervision:** Samuel Wanji.

**Writing – original draft:** Lontum B. Ndzeshang, Samuel Wanji.

**Writing – review & editing:** Joseph D. Turner, Samuel Wanji.

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
