## [Decision Letter · Decision Letter 0]

5 Apr 2020

Dear Dr Turner,

Thank you very much for submitting your manuscript "Generation of Loa loa infective larvae by experimental infection of the vector, Chrysops silacea" for consideration at PLOS Neglected Tropical Diseases. As with all papers reviewed by the journal, your manuscript was reviewed by members of the editorial board and by several independent reviewers. The reviewers appreciated the attention to an important topic. Based on the reviews, we are likely to accept this manuscript for publication, providing that you modify the manuscript according to the review recommendations. 

Sincerely,

Edward Mitre

Guest Editor

Sara Lustigman

Deputy Editor

Reviewer's Responses to Questions

**Key Review Criteria Required for Acceptance?**

**Methods**

-Are the objectives of the study clearly articulated with a clear testable hypothesis stated?

-Is the study design appropriate to address the stated objectives?

-Is the population clearly described and appropriate for the hypothesis being tested?

-Is the sample size sufficient to ensure adequate power to address the hypothesis being tested?

-Were correct statistical analysis used to support conclusions?

-Are there concerns about ethical or regulatory requirements being met?

Reviewer #1: The objectives of the study to develop a tractable model for evaluating drug efficacy is clearly articulated. The study design set forth by the authors largely addresses the objectives in testing the hypothesis. No major concerns in the results, but the absence of sham infections is missing. (see comments below).

Reviewer #2: The objectives of the study are clearly articulated, the study design is appropriate. The sample size if acceptable. Statistical analysis requires minor revision. 

I understand that the ethics and protocol were approved and received clearances from the a committee dvoted to the Institution where the work was conducted (REFOTDE). I don't know if it was done this way because there is no alternative in the country or for other reasons. Some clarifications (in the cover letter at least) would be informative to readers unfamiliar with the regulation in the country where it was done (Republic of Cameroon).

**Results**

-Does the analysis presented match the analysis plan?

-Are the results clearly and completely presented?

-Are the figures (Tables, Images) of sufficient quality for clarity?

Reviewer #1: Yes, the analysis is fine.

Some of the figures (images) need better clarity. (see comments below)

Reviewer #2: The analysis matches with the analysis plan. However, I think the presentation of the results should be improved.

Table 2: the first four columns are identical to those of table 1. I suggest merging the two tables to avoid redundancy.

It would be interesting to know, for each group, the percentage of chrysops with L3s.

Line 332: I can't figure out where the denominator (2453) comes from. Please clarify.

Line 335: Cuzick test for trend is more appropriate than a Chi2 test.

Lines 347-351: this should be in the method section.

In table 5: how many L3 were injected per mice? This should be given as supplementary material.

**Conclusions**

-Are the conclusions supported by the data presented?

-Are the limitations of analysis clearly described?

-Do the authors discuss how these data can be helpful to advance our understanding of the topic under study?

-Is public health relevance addressed?

Reviewer #1: Yes

Reviewer #2: The conclusion is supported by the data presented. Some limitations are addressed. A section addressing the cost could be interesting. 

Lines 419-421: this statement should be supported by data.

**Editorial and Data Presentation Modifications?**

Reviewer #1: (No Response)

Reviewer #2: See above for suggestions concerning the results section.

In addition:

Line 250: add "." after giving the 2 refs.

Line 92: I suggest adding "during" between "post-ivermectin" and "mass treatment".

Lines 95-99: the sentence is really (too) long, the authors should split it. In addition, the new WHO roadmap now targets onchocerciasis elimination by 2030.

Line 145: "were" seems missing before "undertaken".

Line 255: ref 28 is cited before refs 26 and 27.

Discussion:

Lines 382 - 391: this part is redundant with the introduction. It is generic. The authors should go up to the point of their own work.

**Summary and General Comments**

Reviewer #1: The availability of a tractable model to test/evaluate drug candidate efficacies is highly needed, and the authors present a workable model.

No major comments.

Because the authors kind of rely on the yield/death of the flies to finally go forward with the 100 mf condition, why were sham infections not carried out? 

Figure 1. Does not help. A more clear and close-up view would probably do more justice.

Figure 3. Again a better resolution will be helpful.

Figure 4g. Cannot make out the caudal papillae that the camouflaged black arrows are pointing to in the dark green background.

Ln 114. Tentative experiments - Not sure if this would apply to studies that have already been done..!! 

Methods: Lns 138-144. This is not needed in the methodology section. It is anyway mentioned again later.

Ln 206 - Should read as 14 days post infection

Ln 244 - 'eutheised by rising CO2': should it be euthanized with CO2?

Reviewer #2: The article entitled "Generation of Loa loa infective larvae by experimental infection of the vector,

 Chrysops silacea" presents the development of a semi-experimental model of Loa loa lifecycle.

This is a very good piece of parasitology research.

PLOS authors have the option to publish the peer review history of their article (what does this mean?). If published, this will include your full peer review and any attached files.

Reviewer #1: No

Reviewer #2: No
---

## [Editor Report · Decision Letter 1]

22 May 2020

Dear Dr Turner,

We are pleased to inform you that your manuscript 'Generation of Loa loa infective larvae by experimental infection of the vector, Chrysops silacea' has been provisionally accepted for publication in PLOS Neglected Tropical Diseases.

Best regards,

Edward Mitre

Guest Editor

Sara Lustigman

Deputy Editor

---

## [Editor Report · Acceptance letter]

7 Aug 2020

Dear Dr Turner,

We are delighted to inform you that your manuscript, "Generation of Loa loa infective larvae by experimental infection of the vector, Chrysops silacea," has been formally accepted for publication in PLOS Neglected Tropical Diseases.

Best regards,

Shaden Kamhawi

co-Editor-in-Chief

Paul Brindley

co-Editor-in-Chief
